# Hemp Seed (*Cannabis sativa* L.) Varieties: Lipids Profile and Antioxidant Capacity for Monogastric Nutrition

**DOI:** 10.3390/ani14182699

**Published:** 2024-09-18

**Authors:** Elena Rosso, Rosangela Armone, Annalisa Costale, Giorgia Meineri, Biagina Chiofalo

**Affiliations:** 1Department of Science and Pharmaceutical Technology, University of Turin, 10125 Turin, Italy; elena.rosso@unito.it; 2Department of Veterinary Sciences, University of Messina, 98168 Messina, Italy; rosangela.armone@studenti.unime.it (R.A.); biagina.chiofalo@unime.it (B.C.); 3Department of Veterinary Sciences, School of Agriculture and Veterinary Medicine, University of Turin, 10095 Grugliasco, Italy; giorgia.meineri@unito.it

**Keywords:** hemp seeds, fatty acids, antioxidant activity, phenolic compounds, animal feed

## Abstract

**Simple Summary:**

The European Union is working towards a sustainable future by implementing strategies to guide the agricultural transition towards fairer, healthier, and more environmentally friendly food systems, increasing their profitability and sustainability. As a result, food producers are looking for different and stronger products. (*Cannabis sativa* L.), due to its distinctive nutritional profile, can be considered an interesting and promising alternative resource for agriculture in human and animal nutrition. In this study, the proximate composition, fatty acid profile, antioxidant activity, total phenolic and N-*trans*-Caffeoyltyramine content were studied on three hemp varieties (*Carmaenecta*, *Enectaliana*, and *Enectarol*) grown in Central Italy. *Enectarol* had the highest total lipid content and the best antioxidant activity; *Carmaenecta* showed the best fatty acid profile and nutritional indices, and *Enectaliana* showed the highest crude protein and dietary fiber content. All the varieties revealed a high content of essential fatty acids (linoleic acid and α-linolenic acid) and oleic acid. In conclusion, the results highlight that hemp seeds can be used in the food industry as a source of oil and protein and as a supplement in feed mixtures for the valuable antioxidant activity and fatty acid profile, promoting better health in farm animals.

**Abstract:**

The present research aimed to study the proximate composition, fatty acid profile, antiox-idant activity, total phenolic and N-*trans*-Caffeoyltyramine content of three distinct varieties of hemp seeds (*Carmaenecta*, *Enectaliana* and *Enectarol*, grown in a Mediterranean area (Central Italy), as feed in the diet of farm animals. Proximate composition was determined using the official methods of analyses; the fatty acid profile was determined by gas chromatography, total phenolic content (TPC) and the scavenging activity (DPPH^•^ and ABTS^•+^) by the colorimetric method, and N-*trans*-Caffeoyltyramine content by HPLC analysis. The hemp seed *Enectarol* showed the highest total lipid content and the best antioxidant activity with the highest TPC, N-*trans*-Caffeoyltyramine content, and ABTS^•+^, and the lowest peroxidation index and DPPH^•^; *Carmaenecta* showed the best fatty acid profile and nutritional indices (atherogenic and thrombogenic indices and hypocholesterolemic/hypercholesterolemic ratio), and *Enectaliana* showed the highest crude protein and dietary fiber content. The differences observed in the chemical composition, fatty acid profile and antioxidant activity are because of the varieties, considering that all other growing conditions were the same. The results obtained suggest that hemp seed can be used as a source of lipid and protein in animal diets due to their valuable antioxidant activity and as a rich source of essential fatty acids.

## 1. Introduction

The high environmental impact of the food sector has been much debated in recent years. Their production, processing, transport, the area devoted to fodder crops and the considerable consumption of water needed for their irrigation has an impact on the world’s population, with a constant increase in food needs and production, and a further impact on climate change [1,2]. The European Union (EU) is working for a sustainable future through the implementation of strategies such as the Green Deal and One Health, and Farm to Fork (F2F), the heart of the European Green Deal, a 10-year plan created by the European Commission to guide the agricultural transition towards fairer, healthier and more environmentally friendly food systems, increasing their profitability and sustainability [3]. Food producers are therefore exploring alternative and resilient crops. In this context, agricultural hemp (*Cannabis sativa* L.), with a distinctive nutritional profile, could be represent an interesting and promising alternative source for agriculture in human and animal nutrition. Hemp, a dicotyledonous, annual, herbaceous, angiosperm plant, is considered a low environmental impact plant, as it does not require large amounts of water or the use of pesticides and acts as an antagonist to weeds [4]. At the same time, as well as being able to grow rapidly in different agroecological conditions, it is an excellent candidate for carbon sequestration due to its height (it can reach up to four meters) [4,5,6]. The introduction of hemp-based products (hemp seed (HS), hemp cake, hemp seed oil, hemp flour and hemp fiber) as feed material is regulated by Regulation 2022/1104 [7]. Hemp varieties cultivated as ingredients in animal feed must be listed in the European Union (EU) Common Catalogue of Varieties of Agricultural Plant Species, with the maximum content of delta-9-tetrahydrocannabinol (THC), which is the main psychoactive substance, limited to 0.2% (*w*/*w*, in dry matter basis) [7].

As regards the chemical composition [8], the HS is an excellent protein (on average 25% on dry matter, DM) and lipid source in animal feeding (on average 30.9% DM). Lipids are a rich source of essential fatty acids [9], containing up to 80% polyunsaturated fatty acids (PUFA), in which linoleic acid (LA, C18:2 n-6) and α-linolenic acid (ALA, C18:3 n-3) contents are as high as 60 and 19%, respectively [10] compared with other vegetable oils, with the exception of linseed oil [11]. In addition, intermediate fatty acids such as γ-linolenic acid (GLA, C18:3n6) and stearidonic acid (SDA, C18:4n3) [12] have been found in HS oil independently from the precursor fatty acids. The presence of these long-chain polyunsaturated fatty acids (LCPUFA) is a unique characteristic for HS because these fatty acids are not obtained in any other common industrial oilseed crop. LCPUFAs play an important role in immune system regulation, blood clotting, neurotransmitters, cholesterol metabolism and the structure of membrane phospholipids in the brain and the retina [13].

Unlike ruminants, the fiber content of whole hemp seed in the diet of monogastrics could be detrimental for their performance and production. The lack of endogenous enzymes for the degradation of hemp structural carbohydrates in their digestive system can reduce feed intake, digestible energy, and the digestibility of nutrients. However, it is also important to take into account that the consumption of dietary fiber provides several health benefits in the animal body. Dietary fiber, including the insoluble fraction, is considered a functional product that acts as a probiotic. In particular, it has been shown to improve insulin sensitivity and to lower total blood cholesterol and low-density lipoproteins (LDL); furthermore, since dietary fiber resists digestion in the small intestine, it reaches the large intestine, where it is fermented by the intestinal microbiota, to produce short-chain fatty acids that act as prebiotics for the microbiota, impacting microbial metabolism and supporting digestive health [14].

Moreover, many studies reported that HS extracts possess strong antioxidant effects [15] as they contain a significant amount of antioxidants, such as phenolic compounds, particularly flavonoids like flavanones, flavanols, and isoflavones, tocopherols, and phytosterols and bioactive compounds, including antioxidative peptides, N-*trans*-Caffeoyltyramine, cannabisin A, and cannabisin B [16,17,18], which exhibit a spectrum of biological activities, ranging from antioxidant and anti-inflammatory characteristics to anti-tumor and anti-neuroinflammatory effects [19,20,21].

Numerous studies have already been carried out that highlighted the amino acid profile, fatty acid profile, antioxidant content, antioxidant status of egg yolk, milk and meat provided by animals through the dietary incorporation of hemp seed, hemp-seed cake, meal hemp seed or oil hemp seed [22,23]. However, their nutritional and especially functional profile is still poorly characterized. This is probably due to the fact that hemp cultivation is still small-scale, even though Europe, as reported by Horne [24], is one of the main countries in terms of hemp production area. Furthermore, as reported by Ely and Fike [25], this is because the various hemp industries are in their early stages, without significant markets capable of absorbing large quantities of hemp scraps and consequently generating large volumes of co-products.

With this in mind, the aim of the study was to evaluate the chemical and fatty acid composition, total phenolic content, antioxidant activity and N-*trans*-Caffeoyltyramine, as the major component with antioxidant activity, in three hemp seed varieties (var. *Carmaenecta*, *Enectaliana*, *Enectarol*), offering insights into their nutritional, antioxidant and bioactive potential for animal nutrition.

The seeds of these three plant varieties were chosen on the basis of the different characteristics expressed by the plants themselves; the two cultivars *Carmaenecta* and *Enectaliana* have flowers rich in cannabidiol while the cultivar *Enectarol* produces flowers rich in cannabigerol, in order to highlight any differences in composition and activity already present in the seeds. The general objective is to harmonize animal diets in line with the specific needs of the various animal species and their different metabolic pathways.

## 2. Materials and Methods

### 2.1. Chemicals and Plant Material Samples

Folin–Ciocalteu phenol reagent (FCR), gallic acid, 2,2-diphenyl-1-picrylhydrazyl (DPPH^•^), 2,2-azinobis (3-ethyl-benzothiazoline-6-sulfonic acid) (ABTS^•+^), potassium persulphate, 6-hydroxy-2,5,7,8-tetramethy-chromam-2-carboxylic acid (Trolox), sodium carbonate, ethanol (HPLC grade), and acetonitrile (HPLC grade), was purchased from Sigma-Aldrich (Sigma-Aldrich, Milan, Italy). Trifluoroacetic acid (TFA) was obtained from Merck KGaA (Darmstadt, Germany).

Cannabidiol-calibrated solution in ethanol (50 mg/mL, 99.9%), (−)-*trans*-Δ9-THC-calibrated solution in ethanol (100 mg/mL, 99.7%) and Cannabidiolic-acid-calibrated solution in ethanol (1 mg/mL) were purchased from Lipomed AG (Arlesheim, Switzerland).

Solvents and chemicals were all analytical grade.

Hemp (*Cannabis sativa* L.) seeds of the varieties *Carmaenecta*, *Enectaliana*, and *Enectarol* are newly developed industrial hemp cultivars designed for maximum yield, ideal for outdoor cultivation. These varieties are EU-certified and comply with current legal regulations, containing THC levels below 0.2%, as stipulated in Commission Regulation (EU) No 2022/1104 [7]. The seeds were kindly provided by Enecta BV, www.enecta.it (accessed on 15 September 2024). *Cannabis* plants were sown in a low-textured loamy soil in Central Italy (Abruzzo, 42.157402, 13.72528) with low pH (5–6), low salinity (0.8 mS/cm), average organic matter (3–4%), medium nitrogen 100–150 ppm. The plots were 7.000 plants per hectare. Sowing was executed on 15 May 2021 and there was a water dripping system for each row of plants.

Seeds were harvested between September and October, according to the physiological maturity of genotype.

The seeds are collected from a plot of 2 ha (20,000 m^2^). The plants were grown in the same area in a randomized block design with 3 replications with a subplot of 9 m^2^ (3 m × 3 m).

Weed control was performed each month until July, when the plants were tall enough to create shadows that did not allow weeds to grow. After sowing, the air temperature increased from 16 °C to 35 °C in August. Subsequently, the air temperature gradually decreased and reached a minimum of 13 °C in October. The season was a little dry, there was rainfall only at the end of September–October during the harvest.

### 2.2. Sample Preparation

Before all analyses, seeds of each variety, taken from different subplot, were all finely ground together with a 1.1 mm sieve by a grounding machine (Mockmill 200, Wolggang Monck GmbH, Otzberg, Germany) and analyzed. Each analytical determination was replicated three times, and the results expressed as fed.

### 2.3. Proximate Chemical Analysis

Proximate chemical analyses of the HS samples were carried according to the standard procedures from the Association of Official Analytical Chemists [26]: for moisture (method n. 930.15), for ash (method n. 942.05), for crude protein (method n. 2001.11) by means of the Kjeldahl procedure using a Kjeltec system (FOSS, Padua, Italy), for crude fiber (method n. 978.10) using a Fibertec™ 2010 (FOSS, Padua, Italy), and for lipid content (method n. 920.39) by using the Soxtec™ 8000 Extraction Systems and Hydrotec™ 8000 Hydrolysis Systems supplied from FOSS (Padua, Italy). Before the lipid extraction, each sample was hydrolyzed with HCl (3 N) in the Hydrotec system. A Total Starch Assay Kit (Megazyme©, NEOGEN, Lansing, MI, USA) for the analysis of total starch was used, employing AOAC method no. 996.1 [26], involving enzymatic hydrolysis with α-amylase and amyloglucosidase. A spectrophotometric quantification was carried out using a UV-2600 (UV–visible spectrophotometer) from Shimadzu (Milan, Italy). The absorbance was measured at a wavelength of 510 nm. Insoluble, soluble, and total dietary fiber (IDF, SDF, and TDF, respectively) content was determined according to Association of Official Analytical Chemists [26] method 991.43 by means of a Megazyme Total Dietary Fiber Assay Kit (Megazyme©, NEOGEN, Lansing, MI, USA), involving enzymatic hydrolysis with α-amylase, protease and amyloglucosidase, and calculated as the weight of residue minus the weight of protein and ash.

### 2.4. Analysis of Fatty Acids, and Calculation of Nutritional and Quality Indices

Lipid extracts of HS were analyzed after oil extraction and their conversion into fatty acid methyl esters (FAMEs) according to the Christie method [27]. FAMEs were analyzed by a gas chromatography system (GC-FID, TRACE 1310, Thermo Fisher Scientific, Milan, Italy). For FAME separation, an Omegawax 250 capillary column (30 m × 0.25 mm i.d. × 0.25 µm film thickness; Supelco, Bellefonte, PA, USA) was used, as previously described by Oteri et al. [28]. Peaks were identified by comparing the relative retention times of FAME identified in the sample with the retention times of the certified standard mixture (mix 37 FAMEs, Supelco, Inc., Bellefonte, PA, USA) analyzed using the same chromatographic method. ChromeleonTM Data System Software (Version 7.2.9, Thermo Fisher Scientific, Milan, Italy) was employed for GC-FID data collection.

The single FA concentration was expressed in g/100 g, where 100 g is the total of all areas of the identified FAMEs. The atherogenic (AI) and thrombogenic (TI) nutritional indices [29], the hypocholesterolemic/hypercholesterolemic ratio (H/H) [30], and the peroxidation index (PI) [31] were calculated.

### 2.5. Extraction Procedures

Referring to the recently published scientific studies [28,32], improvements were made, thus obtaining the following procedure.

The dried seeds were finely grounded and then suspended in ethanol (solvent/matrix, 30 mL/5 g), sonicated for 10 min in a US-bath (Elmasonic S10H, Elma Schmidbauer GmbH, Singen, Germany) operating at 20 KHz, and then left under reflux for 2.5 h. After filtration with a filter paper, grade 1, the obtained green liquid extract was diluted with ethanol until it reached a volume of 50 mL.

### 2.6. Total Phenolic and Antioxidant Activity Analyses

The extract, diluted 1:10, was used for the determination of the total phenolic content (TPC) and the antioxidant activity.

Using the Folin–Ciocalteu colorimetric technique [16], the total phenolic content (TPC) was determined as follows: 500 µL of a 10% sodium carbonate solution, 250 µL of ethanolic extract, 250 µL of FCR, and 4 mL of water were combined. After centrifuging the mixture for five minutes at 5000 rpm and incubating it for 25 min at room temperature in the dark, the mixture’s absorbance was measured at a wavelength of 765 nm using a Cary 60 UV–Vis spectrophotometer (Agilent Technologies, Santa Clara, CA, USA). Gallic acid equivalents (GAE, mg/g) were used to express the results in terms of mg per gram of material. Three copies of each determination were made.

By using the DPPH^•^ and ABTS^•+^ assays, polyphenol ethanol extracts were evaluated for their ability to act as antioxidants. According to the method outlined by Brand-Williams et al. [32], the DPPH^•^’s scavenging activity was assessed.

The DPPH radical is stable in ethanol solution, antioxidant extracts scavenge the DPPH^•^, and a drop in absorbance at wavelength 517 nm indicates that the DPPH radical has been reduced. After mixing 0.25 mL of 1 mM DPPH^•^ and 2 mL of ethanol with an aliquot of ethanol extract (0.1 mL) containing various quantities (ranging from 2 to 10 mg/mL of the initial sample), the absorbance was measured after 20 min. The amount (µg) of polyphenols required to scavenge 50% of the DPPH was calculated using data (EC50). Then, EC50 values, defined as the amount of dried extract (in mg/mL solution) required to scavenge 50% of initial DPPH, were assessed. The technique described by Oteri et al. [28] was used to assess the Trolox equivalent antioxidant capacity (TEAC). A total of 20 µL of ethanolic Trolox standard solution or pure ethanol were combined with 2 mL of the radical cation from the ABTS^•+^ diammonium salt, vortexed, and heated to 30 °C. Within 6 min, the absorbance was read at = 734 nm. Trolox equivalents (mol TE)/g of seed were used to express the results.

### 2.7. HPLC Analysis of Phenolic Compound N-trans-Caffeoyltyramine

Two different instruments were used to perform this type of analysis. The N-*trans*-Caffeoyltyramine present was quantified using an HPLC-DAD, while its identification was performed with an HPLC-MS PDA system (Waters Corp., Milford, CT, USA). HPLC-MS analyses were carried out on a Waters Fraction Link autopurification system equipped with a Waters 2487 UV detector, Waters 2525 binary pump, and Waters Micromass ZQ detector operating in ESI+ mode. The column used was a Zorbax Eclipse C8 XDB column (150 mm, 4.6 mm, 5 μm; Agilent, Santa Clara, CA, USA), the mobile phase was water (A) and MeCN (B), both with 0.1% HCOOH. The method used included a gradient and started from 2% B, which was maintained for 6 min, up to 12.5% B over the 6–23 min period, followed by: a 30% B step at 23–33 min, a 45% B step at 33–38 min, a 25% B step at 38–42 min, and finally a 100% B step at 42–47 min. The flow rate used was 1.0 mL/min. The injection volume was 1 μL. The main parameters set for the mass spectrometer were: capillary voltage, 2.5 kV; cone voltage, 20 V; cone gas flow, 50 L/h; source temperature, 110 °C; desolvation temperature, 220 °C; desolvation gas, nitrogen; flow rate, 500 L/h; data acquisition range, *m*/*z* 100–1000 Da; and a positive ionization mode.

HPLC-DAD analyses were conducted on a Waters 1525 pump linked to a 2998 PDA (Waters Corp., Milford, CT, USA), using the same column and the same method for HPLC-MS analyses and reported above. UV spectra were recorded at 319 nm, while three-dimensional data were acquired in the 200–400 nm range.

The chromatograms obtained (Figure 1) consist of several peaks, and identification of the N-*trans*-Caffeoyltyramine peak was carried out by comparing mass and UV spectra found in the literature [33] and the spectrum of the standard used.

Quantitation was performed using a calibration line with an external standard. The amount of N-*trans*-Caffeoyltyramine was expressed as mg/g seeds.

### 2.8. Statistical Evaluation

The results obtained were analyzed with the one-way ANOVA test, using the Shapiro–Wilk test to verify the normality and the Levene’s test to verify the assumption of homogeneity of variances.

Tukey’s test (*p* ≤ 0.05) was used for mean separation.

Coefficients were considered significant at *p* ≤ 0.05 (R studio, Posit Software 3.6.3, PBC).

## 3. Results

### 3.1. Nutritional Composition of Hemp Seeds

Table 1 shows the mean values of the chemical composition in the three varieties of HS. *Enectarol* had significantly (*p* < 0.05) higher total lipid content and the lowest crude protein and ash content. *Carmaenecta* showed significantly (*p* < 0.05) higher starch and ash content and the lowest crude fiber, total and insoluble dietary fiber. *Enectaliana* showed significantly (*p* < 0.05) higher crude protein, crude fiber, and total dietary fiber (TDF), and both soluble (SDF) and insoluble (IDF) fractions.

### 3.2. Fatty Acids’ Profiles, Nutritional, and Quality Indices

The fatty acid composition in the three genotypes of HS is presented in Table 2. In descending order, the linoleic (LA, C18:2n6), α-linolenic (ALA, C18:3n3), oleic (C18:1n9), palmitic (C16:0), and stearic (C18:0) acids were the dominant fatty acids in all samples. The linoleic acid showed a significantly (*p* < 0.05) higher value in the *Carmaenecta*. The linolenic acid was the highest (*p* < 0.05) in *Enectaliana* and the lowest (*p* < 0.05) in *Enectarol* and *Carmaenecta*. The oleic acid was significantly (*p* < 0.05) higher in *Enectarol* compared to the remaining genotypes. Palmitic and stearic acids showed significantly (*p* < 0.05) higher values in *Enectaliana*. The γ-linolenic acid (GLA, C18:3n6) was significantly (*p* < 0.05) higher in *Enectaliana* compared to the remaining varieties. The erucic acid (C22:1n9) showed similar values in the three genotypes of HS.

Fatty acid classes, fatty acid ratios, and atherogenic (AI), thrombogenic (TI) and peroxidation (PI) indices are shown in Table 3. *Enectaliana* showed a significantly higher value of saturated fatty acids (SFA). *Enectarol* showed a significantly higher value of monounsaturated fatty acids (MUFA). Polyunsaturated fatty acids (PUFA) and n6-PUFA were higher in *Carmaenecta* and *Enectaliana* than the remaining genotype. *Enectaliana* showed the highest value of n3-PUFA. *Enectaliana* also showed the highest SFA/UFA ratio, while *Enectarol* and *Carmaenecta* showed the lowest one. *Carmaenecta* showed also the lowest SFA/UFA and n3/n6 PUFA ratios. With regard to the nutritional indices, strictly related to the fatty acid profile, *Enectaliana* showed the worst AI, while no significant differences were observed for TI among the three varieties. *Enectarol* showed the best hypocholesterolemic/hypercholesterolemic ratio and the lowest PI. The remaining genotypes showed significantly higher and not statistically different PI values.

### 3.3. Total Phenolic Content and Antioxidant Activity

Total phenolic content, antioxidant properties, and phenolic compounds in the three varieties of hemp seeds are shown in Table 4.

The ER variety has the highest values of antioxidant activity for all test types among the three studied, followed by the EC variety, which has intermediate antioxidant activity, and the EL variety, which has the lowest antioxidant activity.

Regarding the results of the DPPH test and the amount of N-*trans*-Caffeoyltyramine found, the difference between the seeds of the ER and EC varieties was not statistically significant. For both, however, the values obtained were the highest in the study.

## 4. Discussion

In the present study, HS varieties showed an interesting nutritional profile, containing large amounts of crude protein, oil and crude fiber at 24–25, 26–29 and 21–26 g/100 g as fed, respectively. Similar values for crude protein were reported by EFSA [8], Alonso-Esteban et al. [34], House et al. [35], Callaway [36] and Klir et al. [37] testifying that, among other protein sources largely diffused in animal feeding, the HS can be located as an intermediate crude protein (CP) source, between soybean (on average 39.2 g/100 g, on DM) and sunflower seeds (on average 19.2 g/100, on DM) [38]. The high average total lipid content is in line with that reported in the literature [8,35,37,39], but lower than that reported by Callaway [36]. Starch content in the three varieties of hemp seeds resulted higher than that reported by Fan et al. [40]. Crude fiber content was similar to that reported by Razmaitè et al. [39] and lower than that reported by Callaway [36] and House et al. [35] and EFSA [8]. Insoluble fibers (IDF) are the dominant fibers in HS, consisting of approximately cellulose (46%), lignin (31%), and hemicellulose (22%) [41]. The values of IDF and SDF were higher than those reported by Fan et al. [40]. The consumption of whole oil crop seeds may have an impact on the intake of fiber. As the dietary fiber of HS remains highly fermentable [40], the potential use of HS as a functional ingredient, capable of exerting an effect on the gut microbiota and its metabolites, has been demonstrated [42,43].

An interesting composition of fatty acids is evident in HS. Among the fatty acids of nutritional interest, significant quantities of linoleic acid (LA), which accounts for more than half of the total fatty acids (53–55%), approximately 18% of α-linolenic acid (ALA) and approximately 15% of oleic acid, were observed; γ-linolenic acid (GLA) ranged from 0.5% (*Enectarol*) to 1.8% (*Enectaliana*). Our results are similar to those reported by Czerwonka and Bialek [44], and higher than those reported by Alonso-Esteban et al. [34], except for GLA, which showed lower values. The LA content in hemp seeds is similar to that of corn, cottonseed and soybean, and higher than that of flaxseed, safflower, olive oil and palm; the ALA content is lower than that of chia and flaxseed but higher than of the most common seeds used as ingredients for animal feeding [39,44].

Some genera of the Cannabaceae family contain GLA, but among the plant sources used as nutraceuticals in animal nutrition, the most important are evening primrose (*Oenothera biennis* L.), blackcurrant (*Ribes nigrum* L.) and borage (*Borago officinalis* L.), which contain GLA, respectively, 9–12%, 15–20% and 21–23% of total fatty acids [45].

In agreement with the results of Czerwonka and Bialek [44], HSs of all varieties showed low percentages of SFA and MUFA, while large amounts of PUFA rich in both n6- (54–56%) and n3 -PUFA (18%) were observed, with a n6/n3 PUFA ratio (3:1) optimal for immune balance, for the synthesis of eicosanoids, and to reduce the propensity to inflammation [46]. The seeds tested showed a high atherogenic value (AI) due to their high content of unsaturated fatty acids and thrombogenic (TI) indices and the hypocholesterolemic/hypercholesterolemic ratio (H/H), compared to amaranth, quinoa and pumpkin seeds, while the nutritional indices of chia and flax were similar to those of hemp seeds [44]. Among the nutritional indices, values similar to those reported by Razmaitè et al. [39] were observed for AI and TI, and for the n6/n3 PUFA ratio to those observed by Jing et al. [47], Benkirane et al. [48] and Taaifi et al. [49]. HS are neither non-atherogenic nor thrombogenic and have a very high hypocholesterolemic effect (H/H), as shown by the low lipid quality indices (AI and TI). AI, TI and H/H are used to evaluate the nutritional value of lipids; low AI and TI and high H/H show to the potentially beneficial effect of lipids for the functioning of the cardiovascular system [50]. The PI index of all HS varieties was similar to that indicated by Kang et al. [51], highlighting only in the *Enectarol* variety the value included in the most favorable range (80–90) to reduce the risk of cardiovascular diseases [51].

Concerning healthy fatty acids, hemp seed, thanks to its low SFA and high PUFA content offers positive health benefits such as lipid metabolism, cardiovascular health, immunomodulatory effects and dermatological diseases [52]. LA and ALA are essential fatty acids and represent the substrate for the synthesis of bioactive very long-chain fatty acids such as, respectively, GLA and arachidonic acid (AA, C20:4n6) for the n6 series, and eicosapentaenoic (EPA, C20:5n3), docosapentaenoic (DPA, C22:5n3) and docosahexaenoic (DHA, C22:6n3) for the n3 series precursors in the synthesis of eicosanoids (prostaglandins, thromboxanes and leukotrienes) which are modulators of the immune-inflammatory system [53,54].

Although EFSA [8] suggests that no tolerable upper intake level should be set for total or individual n-6 PUFAs, there is research evidence [54] to suggest that a reduction in the dietary intake of LA and ARA, together with an increase in n-3 long-chain PUFAs, would be beneficial for most consumers. In particular, among the fatty acids of the n6 series, GLA is a metabolic intermediate of the biosynthetic pathway which, in addition to modulating the immune–inflammatory system [55], is also useful for the treatment of several skin disorders [56]. GLA synthesis is low and rate-limiting; therefore, the dietary contribution of GLA helps to reduce the possible deficiency of this fatty acid in the human and animal body [57].

It is known that PUFA, and particularly ALA, are very susceptible to oxidation [58], which increases the risk of oil damage. Although in the present study the PI index indicated the poor oxidative stability of oils, it is known that whole oil crop seeds are chemically stable. The consumption of whole oil crop seeds, such as hemp, flax, and camelina, can help to avoid oil deterioration and, consequently, detrimental health effects [39].

Scientific research is increasingly investigating the properties of phenolic compounds due to their associated benefits, which can be exploited in the formulation of functional food and nutraceuticals [59]. The results of this study are similar to those reported in the literature; in particular, the total phenol content is comparable to that reported by Benkirane et al. [33], studying the differences between two seed varieties collected from four different Moroccan regions and with Irakli et al. [60] studying the effect of genotype and growing year on the nutritional, phytochemical, and antioxidant properties of industrial HS. The differences observed in the TPC and N-*trans*-Caffeoyltyramine extracted from HS depended on the different varieties, considering that pedological, climatic situations and agronomic practices were the same [61]. However, it is essential to specify that the types of compounds extracted varies depending on the process used, including the extraction method, solvent, and conditions applied. The choice of solvent and extraction method significantly impacts the final results. As indicated by Kalinowska et al. [62], hemp seed extracts analyzed for phenolic compounds with antioxidant properties, fatty acids, and antioxidant activity are more effectively obtained using methanol and ethanol.

As regards the scavenging activity (DPPH^•^ and ABTS^•+^), the comparison with the results reported by the literature must be very cautious, since there are many assays for total antioxidant determination and each of them has its limitations [63]. In fact, Rotta et al. [64] reported the importance of measuring the antioxidant property using at least two methods based on different reaction mechanisms. Therefore, in this study, the antioxidant capacity was evaluated using the capacity to scavenge the “stable” free radical 2,2 diphenyl-1-picrylhydrazyl (DPPH^•^) [32] and the 2,2′-Azino-bis(3-ethylbenzothiazoline-6-sulfonic acid) diammonium salt radical cation (ABTS^•+^) frequently used to estimate the total antioxidant capacity of natural products, including crude extracts, polyphenols, phenolic acids, flavonoids, and others [65].

The results recorded for the ABTS^•+^ test are comparable to those reported in the literature of Irakli et al. [60], while the results of the DPPH^•^ test are comparable to those reported by Chen et al. [16], who studied forty samples extracted from the kernels and hulls of two varieties of HS, and they are better than those reported by Benkirane et al. [33].

In our study, the TPC values across the three hemp varieties (*Enectarol*, *Carmaenecta*, and *Enectaliana*) were significantly different, with *Enectarol* showing the highest concentration. These phenolic compounds, including N-*trans*-Caffeoyltyramine, are responsible for the strong antioxidant activity observed, particularly in their ability to scavenge free radicals. This was evident from both the DPPH^•^ and ABTS^•+^ assays, where *Enectarol* had the lowest EC50 for DPPH^•^ (indicating a higher scavenging activity) and the highest values for ABTS^•+^ radical scavenging capacity.

Specifically, N-*trans*-Caffeoyltyramine was identified as a major phenolic compound contributing to these antioxidant effects [16]. The compound’s concentration was highest in *Enectarol*, followed closely by *Carmaenecta*, and significantly lower in *Enectaliana*. The superior performance of *Enectarol* in antioxidant tests, coupled with its high TPC and N-*trans*-Caffeoyltyramine content, suggests that this polyphenolic compound plays a key role in the oil’s stability and its potential health benefits. These results are in line with previous findings that emphasize the importance of using multiple assays to comprehensively evaluate antioxidant capacity, as different methods (DPPH^•^, ABTS^•+^) measure varying aspects of radical scavenging [16].

Thus, the enrichment of hemp seeds with polyphenols like N-*trans*-Caffeoyltyramine not only enhances their functional properties as antioxidants but also points to their potential applications in nutraceuticals and functional foods, where they could be used to mitigate oxidative stress and improve health outcomes.

## 5. Conclusions

The results show that hemp seeds can be used in animal diets as a source of oil and protein, as a supplement, and as a rich source of essential fatty acids and phenolic compounds. Due to the high content of unsaturated fatty acids, all the examined hemp seeds have a beneficial index of atherogenicity and thrombogenicity, and a good hypocholesterolemic/hypercholesterolemic ratio. Specifically, the *Enectarol* variety showed the highest total lipid content with the highest MUFA content, especially oleic acid, which is more stable than polyunsaturated fatty acids. Furthermore, this variety showed the best antioxidant activity, with the highest TPC, N-*trans*-Caffeoyltyramine content, and ABTS^•+^, and the lowest peroxidation index and DPPH^•^. The *Carmaenecta* variety showed the best fatty acid profile, with a lower content of SFA and a higher content of PUFA, and the best nutritional indices (AI, TI and H/H). The *Enectaliana* variety showed the highest crude protein and dietary fiber content. The differences observed in the chemical composition, fatty acid profile and antioxidant activity are due to the effect of varieties, considering that the environmental conditions were the same. In conclusion, the results highlight the suitability of all varieties, characterized by the valuable antioxidant activity and fatty acid profiles, as an ingredient in the food industry. However, the *Enectarol* variety could be recommended in the feeding of heavy pigs intended to produce cured products, due to the greater stability of the meat during technological processing, while the *Carmaenecta* variety could be recommended in the feeding of pigs intended to produce fresh meat enriched with fatty acids of nutritional interest. In vivo tests will be necessary to evaluate the effectiveness of these varieties of hemp seeds on the health and growth of farm animals.

## Figures and Tables

**Figure 1 animals-14-02699-f001:**
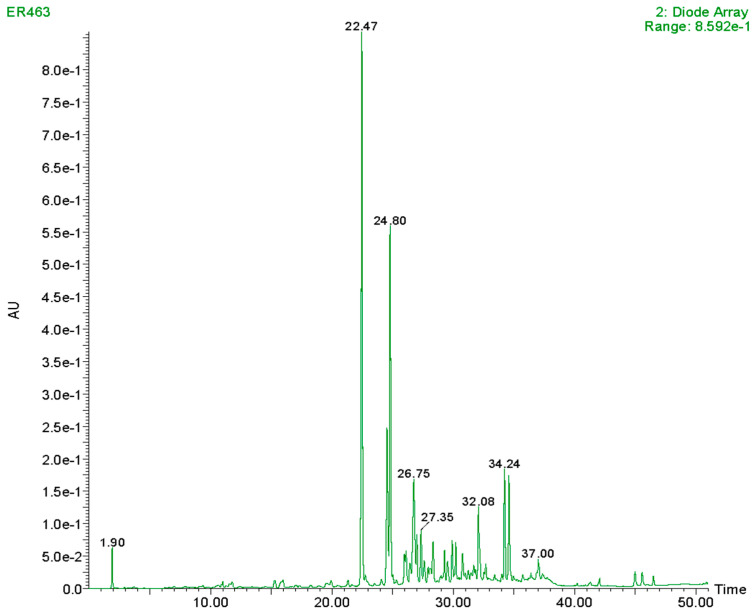
Chromatogram of an ethanolic extract of hemp seed.

**Table 1 animals-14-02699-t001:** Chemical composition (g/100 g, on DM) of the three genotypes of hemp seeds.

		Genotype			
	*Enectarol*	*Carmaenecta*	*Enectaliana*	SEM	Pr(>F)
	(*ER*)	(*CE*)	(*EL*)		
DM	93.16 ^b^	92.10 ^c^	93.46 ^a^	0.016	1.23 × 10^−12^
CP	24.33 ^c^	24.91 ^b^	25.23 ^a^	0.051	1.708 × 10^−6^
Total lipid	30.93 ^a^	30.19 ^b^	30.14 ^b^	0.095	3.708 × 10^−4^
CF	24.42 ^b^	20.98 ^c^	25.58 ^a^	0.088	1.061 × 10^−10^
Ash	3.54 ^c^	5.41 ^a^	5.10 ^b^	0.025	3.464 × 10^−12^
Starch	9.72 ^b^	10.47 ^a^	9.76 ^b^	0.120	2.559 × 10^−3^
TDF	33.12 ^b^	27.57 ^c^	33.92 ^a^	0.120	6.083 × 10^−11^
IDF	31.38 ^a^	26.13 ^b^	31.55 ^a^	0.187	9.118 × 10^−9^
SDF	1.74 ^b^	1.44 ^b^	2.37 ^a^	0.080	5.422 × 10^−5^

DM: dry matter; CP: crude protein; CF: crude fiber; Ash: total mineral content; TDF: total dietary fiber; IDF: insoluble dietary fiber; SDF: soluble dietary fiber. SEM: standard error of mean; Pr(>F) *p* value. Mean values followed by different letters within the same row differ significantly (*p* < 0.05).

**Table 2 animals-14-02699-t002:** Fatty acid profile (g/100 g of total FA, as fed) in the three genotypes of hemp seeds.

	Genotype				
	*Enectarol*	*Carmaenecta*	*Enectaliana*	SEM	Pr(>F)
	(*ER*)	(*CE*)	(*EL*)		
Myristic acid	0.04	0.04	0.04	0.002	0.311
Myristoleic acid	0.03 ^a^	0.01 ^b^	0.03 ^a^	0.003	0.016
Palmitic acid	7.90 ^b^	7.89 ^b^	8.29 ^a^	0.007	5.717 × 10^−3^
Palmitoleic acid	0.15 ^a^	0.11 ^b^	0.11 ^b^	0.006	1.831 × 10^−3^
Heptadecanoic acid	0.05 ^b^	0.06 ^a^	0.06 ^ab^	0.003	0.044
Heptadecenoic acid	0.04 ^a^	0.03 ^b^	0.03 ^b^	0.001	7.128 × 10^−3^
Stearic acid	2.59 ^b^	2.39 ^c^	2.92 ^a^	0.023	1.847 × 10^−7^
Oleic acid	14.97 ^a^	12.77 ^b^	11.45 ^c^	0.118	2.068 × 10^−8^
Vaccenic acid, cis	0.96 ^a^	0.95 ^a^	0.88 ^b^	0.016	9.453 × 10^−3^
Linoleic acid	53.17 ^c^	54.86 ^a^	53.83 ^b^	0.113	8.015 × 10^−6^
γ-Linolenic acid	0.54 ^c^	1.21 ^b^	1.80 ^a^	0.022	6.920 × 10^−11^
α-Linolenic acid	17.56 ^b^	17.79 ^b^	18.44 ^a^	0.140	4.064 × 10^−3^
Arachidic acid	0.79 ^b^	0.78 ^b^	0.86 ^a^	0.013	3.114 × 10^−3^
Eicosaenoic acid	0.38	0.37	0.39	0.008	0.394
Eicosadienoic acid	0.06 ^ab^	0.05 ^b^	0.07 ^a^	0.003	0.019
Dihomo-γ-linolenic acid	0.03 ^b^	0.03 ^b^	0.04 ^a^	0.002	2.061 × 10^−3^
Eicosatrienoic acid	0.03 ^a^	0.01 ^b^	0.02 ^ab^	0.003	0.025
Arachidonic acid	0.04 ^a^	0.01 ^b^	0.01 ^b^	0.003	2.356 × 10^−4^
Behenic acid	0.38 ^ab^	0.36 ^b^	0.39 ^a^	0.006	0.027
Erucic acid	0.05	0.04	0.04	0.003	0.569
Tricosanoic acid	0.07 ^a^	0.06 ^b^	0.07 ^ab^	0.003	0.025
Lignoceric acid	0.20	0.19	0.19	0.006	0.412
Docosaesaenoic acid	0.03 ^b^	0.05 ^a^	0.05 ^a^	0.002	1.574 × 10^−4^
Nervonic acid	0.03 ^b^	0.04 ^ab^	0.05 ^a^	0.003	1.362 × 10^−3^

The concentration of fatty acids was expressed as g/100 g of total FA, considering 100 g the sum of the areas of all FAME identified. C14:0 = myristic acid; C14:1 = myristoleic acid; C16:0 = palmitic acid; C16:1 = palmitoleic acid; C17:0 = heptadecanoic acid; C17:1 = heptadecenoic acid; C18:0 = stearic acid; C18:1n9 = oleic acid; C18:1n7 = cis-vaccenic acid; C18:2n6 = linoleic acid (LA); C18:3n6 = γ-linolenic acid (GLA); C18:3n3 = α-linolenic acid (ALA); C20:0 = arachidic acid;C20:1n9 = eicosaenoic acid; C20:2n6 = eicosadienoic acid; C20:3n6 = dihomo-γ-linolenic acid; C20:3n3 = eicosatrienoic acid; C20:4n6 = arachidonic acid (AA); C22:0 = behenic acid; C22:1n9 = erucic acid; C23:0 = tricosanoic acid; C24:0 = lignoceric acid; C22:6n3 = docosaesaenoic acid; C24:1n9 = nervonic acid. SEM: standard error of mean; Pr(>F) *p* value. Mean values followed by different letters within the same row differ significantly (*p* < 0.05).

**Table 3 animals-14-02699-t003:** Fatty acid classes (g/100 g of total FA, as fed), ratios and quality indices in the three genotypes of hemp seeds.

	Genotype				
	*Enectarol*	*Carmaenecta*	*Enectaliana*	SEM	Pr(>F)
	(*ER*)	(*CE*)	(*EL*)		
SFA	11.99 ^b^	11.75 ^b^	12.79 ^a^	0.102	1.209 × 10^−4^
MUFA	16.59 ^a^	14.33 ^b^	12.97 ^c^	0.140	7.246 × 10^−8^
PUFA	71.43 ^b^	73.92 ^a^	74.25 ^a^	0.237	2.421 × 10^−5^
SFA/UFA	0.14 ^b^	0.13 ^b^	0.15 ^a^	0.001	9.476 × 10^−5^
n3	17.62 ^b^	17.85 ^b^	18.51 ^a^	0.138	3.450 × 10^−3^
n6	53.77 ^b^	56.11 ^a^	55.67 ^a^	0.120	5.504 × 10^−7^
n3/n6	0.33 ^a^	0.32 ^b^	0.33 ^a^	0.002	2.625 × 10^−3^
AI	0.09 ^b^	0.09 ^b^	0.10 ^a^	0.001	2.775 × 10^−3^
TI	0.12 ^b^	0.12 ^b^	0.13 ^a^	0.001	3.946 × 10^−3^
H/H	10.84 ^a^	10.77 ^a^	10.10 ^b^	0.103	9.459 × 10^−4^
PI	89.80 ^b^	93.30 ^a^	94.84 ^a^	0.378	1.833 × 10^−5^
n6/n3	3.05 ^b^	3.14 ^a^	3.01 ^b^	0.019	2.293 × 10^−3^
PUFA/SFA	5.95 ^b^	6.29 ^a^	5.79 ^b^	0.069	2.001 × 10^−3^

The concentration of fatty acids was expressed as g/100 g of total FA, considering 100 g as the sum of the areas of all FAME identified. SFA = saturated fatty acids; MUFA = monounsaturated fatty acids; PUFA = polyunsaturated fatty acids; SFA/UFA = saturated/unsaturated fatty acid ratio; n3 = n3-polyunsaturated fatty acids; n6 = n6-polyunsaturated fatty acids; n3/n6 = n3/n6-polyunsaturated fatty acid ratio; AI = atherogenic index; TI = thrombogenic index; H/H = hypocholesterolemic/hypercholesterolemic ratio; PI = peroxidation index. SEM: standard error of mean; Pr(>F) *p* value. Mean values followed by different letters within the same row differ significantly (*p* < 0.05).

**Table 4 animals-14-02699-t004:** Total phenolic content, antioxidant properties, and phenolic compound in the three varieties of *Cannabis sativa* L. seeds (n = 9).

	Genotype				
	*Enectarol* (*ER*)	*Carmaenecta* (*CE*)	*Enectaliana* (*EL*)	SEM	Pr(>F)
TPC	0.51 ^a^	0.35 ^b^	0.23 ^c^	0.015	4.065 × 10^−5^
DPPH^•^	0.06 ^b^	0.07 ^b^	0.13 ^a^	0.063	3.216 × 10^−4^
ABTS^•+^	4.33 ^a^	2.72 ^b^	1.01 ^c^	0.115	2.927 × 10^−6^
N-*trans*-Caffeoyltyramine	0.2147 ^a^	0.1732 ^a^	0.0655 ^b^	0.022	7.736 × 10^−3^

TPC = Total phenolic content expressed as gallic acid equivalents (GAE mg/g seeds); DPPH^•^ = scavenging activity expressed as EC50 (concentration of dried extract mg/mL solution). ABTS^•+^ = scavenging activity expressed as μmol TE (Trolox equivalents)/g (seeds). N-*trans*-Caffeoyltyramine = phenolic compounds expressed as mg/g seeds. SEM: standard error of mean; Pr(>F) *p* value. Mean values followed by different letters within the same row differ significantly (*p* ≤ 0.05).

## Data Availability

Data are contained within the article.

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
