# Peer review of "Hemp Seed (Cannabis sativa L.) Varieties: Lipids Profile and Antioxidant Capacity for Monogastric Nutrition"

_animals, 2024, doi:10.3390/ani14182699_

Round 1
Reviewer 1 Report
Comments and Suggestions for Authors
Dear authors, please check the comments belowï¼›
KEYWORDS: please remove proximate composition and nutritional indices
INTRODUCTION:
The introduction of the article appears identical to that of Lanzoni et al. (2024) (Phenolic profile and antioxidant activity of hemp co-products following green chemical extraction and ex vivo digestion), without mentioning it even once in the text. For this reason, you are kindly requested to rewrite the introduction, both in line with your experimental objectives and where articles must be cited.
L48: Please modify food sector and especially of feed in food and especially feed sector.
L49-L53: What you have reported is correct, however it is unclear and unrelated to the previous sentence. Please amend.
L62: very low content? Please introduce here the limit (L74). Another requirement for using hemp seeds, apart from their THC content, is that they belong to varieties registered in the European catalogue.
L69-71: This is a scientific opinion, so EFSA has not introduced HS or cake for animal feed. Please specify. The introduction of hemp-based products is regulated by Regulation 2022/1104 (L108).
L76: THC free?
L77: Starch? HS is an excellent lipid and protein source, please modify.
L93: Although it is true that extractions are characterised by a high functional profile, it is essential to specify that the type of compounds extracted is not always the same, but depends on the type of extraction, solvent used, etc.
L107-L108: Not only HSs. Please specify all hemp-based products.
L110: Your idea of characterising 3 different varieties is interesting. But in the introduction you never mention why, what is the reason for using one rather than another.
MATERIALS:
THC content < 0.6%?
METHODS:
Considering that HSs are characterised by a high fibre content, the values of NDF, ADF and ADL must be quantified.
L231: Why only N-trans-caffeoyltyramine?
STATISTICAL EVALUATION
Please remove the parts of Animals template
RESULTS
NDF, ADF and ADL values must be included.
L301: lowest oil content? This is not true, it is highly comparable to Carmaenecta, please correct.
TABLE 2: To facilitate the reading, is better to include the name of FA in the table. Please modify. In addition, change (,) in (.), the same in the TABLE 3
DISCUSSION:
You test 3 different varieties of HSs and your discussion is about hemp seeds in general. I think this should be addressed by reporting references to support what you have observed and the differences between these varieties.
L398: Very high? Please use scientific language
L429: Which seeds? Please specify
L458-460: All the seeds? Certainly if the fat content increases, the stability decreases. But why are some seeds (more than others) more stable? Please explain.
462-510: I agree with mentioning the effects of HSs in in vivo trials, but most of your discussion focuses on this and you spend little discussion on phenolic and antioxidant characterisation, the main topic of your paper (also mentioned in the title). This part should be rewritten.
I believe that tracing antioxidant activity and TPC to a single compound is speculative. Certainly the phenolic content influences antioxidant activity, but the phenolic profile of HSs is broad. Moreover, this part, as previously reported, needs to be further investigated. You have to talk about TPC depending on the extraction you have implemented, which is definitely an influencing factor.
L542-548: This part must be removed not relevant to the discussion.
ABSTRACT & CONCLUSION
Abstract and conclusion must be revised with the new version of the discussion
Comments on the Quality of English LanguageEnglish is well written, only few errors, easily correctable
Reviewer 2 Report
Comments and Suggestions for Authors
L 101-102: Please rephrase this sentence. Numerous studies have already been carried out that highlighted the amino acid profile, fatty acid profile, antioxidant content, antioxidant status of egg yolk, milk and meat provided by animals through the dietary incorporation of hemp seed, hempseed cake, meal hempseed or oil hemp seed. These articles have been published since 2015. I recommend an article that was recently published (2024): Nutraceutical potential, phytochemistry of hemp seed (Cannabis sativa L.) and its application in food and feed: A review; Effect of Dietary Incorporation of Hemp Seeds Alone or with Dried Fruit Pomace on Laying Hens’ Performance and on Lipid Composition and Oxidation Status of Egg Yolks.
L 110-115: If the research was looking at the nutritional potential of hemp seeds, I think it was very important to determine the amino acid profile. The most important problem of animal nutrition is the provision of proteins with high biological value. Hemp products and by-products could be an unconventional source, primarily of protein. The lack of amino acid profiling makes this manuscript too data-poor to be a scientific article.
Table 1: I believe that the term oil is not appropriate, EE (ether extract) should be used because the method used determines the substances soluble in the organic solvent, i.e. fats, pigments, fat-soluble vitamins, resins, etc.
L 561-564: Please specify which of the three cultivars the authors recommend to be incorporated into animal feed. From the conclusions it follows that all three varieties are recommended for the diet of animals - then what are the effects of the conducted study.
Round 2
Reviewer 1 Report
Comments and Suggestions for Authors
Dear authors, thank you for your work.
Below are the comments.
· Typos, abbreviations, spaces and repetition of words should be checked in the text.
ABSTRACT:
L31: Remove (CE), (EL) and (ER).
L 32: Find environment substitute
INTRODUCTION:
L65: Add here HS;
L68: Replace while with “with”;
L69: Is limited. Delete is;
L70: Delete “Seed from industrial hemp is commonly referred to …”
L73: Delete the second on dry matter;
L84-86: Although what you report is true, it is necessary (given the monogastric as the topic of your article) that you also specify the negative effects of a high fibre content in hemp seed on its digestibility. (https://doi.org/10.3390/foods12030601);
L94-99: Move this part to the discussion section;
L117-L118: CBD and CBG, full name;
MATERIALS:
L135: Cannabis sativa in italics. Please also check the rest of the text;
METHODS:
L157: What do you mean?
L267-268: Change figure caption
DISCUSSION:
L439-445: Merge this discussion with the upper part (L414-420);
L470-482: You must expand on the discussion of Caffeoyltyramine with data obtained from TPC and other antioxidant activities. The discussion of these data cannot be limited to a few lines, given the role it plays in your work.
CONCLUSION
L499-L502: This part should be put in the discussions
Comments on the Quality of English Language
Please check the text for typos, abbreviations and repetitions
Author Response
Please see the attachment.
Please check the text for typos, abbreviations and repetitions. Done

Reviewer 2 Report
Comments and Suggestions for Authors
The authors revised and improved the article according to the reviewer's comments and suggestions.
Author Response
The authors revised and improved the article according to the reviewer's comments and suggestions.
Done